# Unlocking Structured Thinking in Language Models with Cognitive prompting

## Abstract

We propose cognitive prompting as a novel approach to guide problem-solving in large language models (LLMs) through structured, human-like cognitive operations such as goal clarification, decomposition, filtering, abstraction, and pattern recognition. By employing systematic, step-by-step reasoning, cognitive prompting enables LLMs to efficiently tackle complex, multi-step tasks. We evaluate the effectiveness of cognitive prompting on Meta's LLaMA models, comparing performance on arithmetic reasoning tasks using the GSM8K dataset and on commonsense reasoning benchmarks. Our analysis includes comparisons between models without cognitive prompting, models with a static sequence of cognitive operations, and models using reflective cognitive prompting, where the LLM dynamically self-selects the sequence of cognitive operations. The results show that cognitive prompting, particularly when dynamically adapted, significantly improves the performance of larger models, such as LLaMA3.1 70B, and enhances their ability to handle multi-step reasoning tasks. This approach also improves interpretability and flexibility, highlighting cognitive prompting as a promising strategy for general-purpose AI reasoning.

## 1 Introduction

Recent advancements in artificial intelligence (AI), especially with large language models (LLMs), have made great progress in emulating human reasoning to solve tasks like text summarization (Stiennon et al., 2020), code generation (Guo et al., 2023), and question answering (Lu et al., 2022). While LLMs excel at generating coherent text and handling vast data, their ability to perform multi-step reasoning still falls short of human cognitive processes. Human cognition, marked by its structured nature, provides a compelling blueprint for guiding AI through complex tasks that require layered thinking and adaptability.

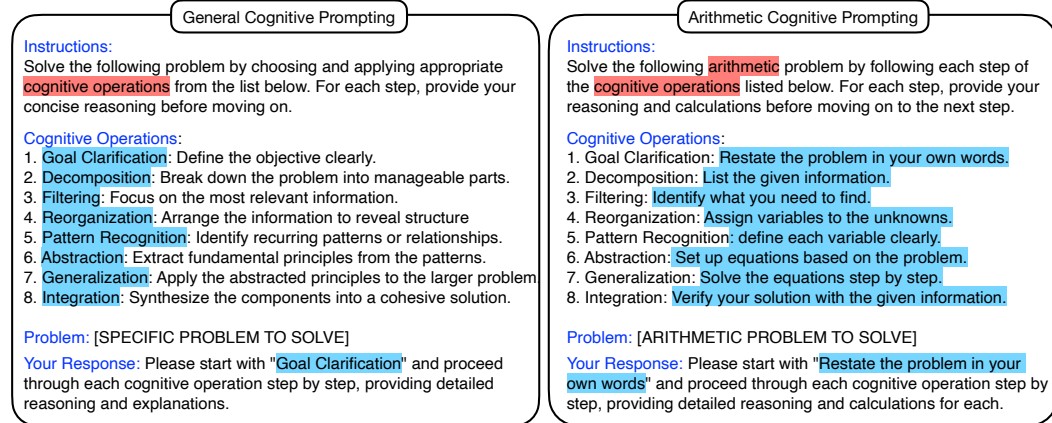

Figure 1: Left: General cognitive prompting, Right: Cognitive prompting adapted to arithmetical reasoning.

This paper introduces a novel approach called cognitive prompting, designed to enhance problem-solving in LLMs by systematically emulating human cognitive operations (COPs). Cognitive prompting organizes problem-solving into distinct cognitive steps—such as goal clarification, task decomposition, and pattern recognition—allowing LLMs to tackle complex tasks in a more structured and interpretable manner, see Figure 1. Inspired by cognitive psychology and cognitive architectures like ACT-R (Anderson & Lebiere, 1996), this method bridges the gap between human-like reasoning and AI's computational power, enabling models to handle tasks in fields such as mathematics, logic, decision-making, and creativity with greater precision. Our experiments, conducted with Meta's LLaMA models (Touvron et al., 2023) on the GSM8K (Cobbe et al., 2021) and a commonsense benchmark (Shi & Lipani, 2024), demonstrate significant improvements in task performance when cognitive prompting is applied. In particular, the reflective variant of cognitive prompting leads to enhanced reasoning capabilities.

The structure of the paper is as follows: Section 2 introduces the concept of cognitive prompting, detailing its core operations and their application in problem-solving. Section 3 presents experimental results on the impact of cognitive prompting on arithmetic reasoning tasks, while Section 4 explores its effectiveness in commonsense reasoning. Section 5 reviews related work on prompting engineering strategies. Finally, Section 6 concludes the paper. The appendix contains exemplary reasoning processes and examples for problem-specific COPs.

## 2 COGNITIVE PROMPTING

Cognitive prompting organizes problem-solving through a structured sequence of human-like COPs, enabling LLMs to tackle complex tasks across domains such as mathematics, logic, creativity, and decision-making. This method, inspired by principles in cognitive psychology, breaks problems into stages like goal clarification, decomposition, filtering, and integration—mimicking the way humans refine their understanding of tasks. By leveraging this structured approach, cognitive prompting enhances clarity, interpretability, and adaptability in LLM reasoning.

Unlike methods like Chain of Thought (CoT) (Wei et al., 2022), cognitive prompting offers more general multi-dimensional operational depth, allowing LLMs to approach a wider variety of problems with reasoning progression. This framework, rooted in dual-process and problem-space theories, encourages both intuitive and analytical reasoning, helping models transition between pattern recognition, abstraction, and integration for more consistent and interpretable solutions. Cognitive prompting can be formalized as an optimization problem. Let $C = \{c_1, c_2, \ldots, c_n\}$ represent a set of COPs and $S = \{s_1, s_2, \ldots, s_k\}$ denote a sequence of $k$ operations from $C$. The objective is to find the sequence $S^*$ that maximizes task performance $S^* = \arg\max_{S \subseteq C} f(S)$ subject to constraints such as $|S| = k$, $s_1 =$ goal clarification, and $s_k =$ integration. Here, $f(S)$ represents task performance, e.g., accuracy, efficiency, coherence.

Cognitive prompting follows a structured process that mirrors human problem-solving. Key COPs include:

**Goal Clarification:** Clearly define the objective of the problem to maintain focus on solving it effectively. In the context of COP, goal clarification ensures that the model aligns its reasoning with the desired outcome, minimizing distractions. Let G represent the goal, and all subsequent operations should be oriented toward achieving G , helping the model concentrate on the correct direction of reasoning.

**Decomposition:** Break down the problem $P$ into smaller, manageable components $\{P_1, P_2, \ldots, P_n\}$, where $P = \bigcup_{i=1}^{n} P_i$. This step is crucial in COP as it allows the model to tackle complex, multi-step problems incrementally. Decomposition is particularly useful in mathematical problem-solving and logic tasks, where breaking a problem into sub-problems allows the model to apply specific operations or strategies to each part. Moreover, decomposition helps to identify the core structure of the problem, isolating the critical steps required for a comprehensive solution.

**Filtering:** Select the most relevant information from $I = \{i_1, i_2, \ldots, i_m\}$ using a filtering function $F(I) = I_{\text{rel}} \subseteq I$. Filtering is essential in COP to prevent the model from being overwhelmed by

irrelevant details. In complex tasks, the problem statement may include redundant or distracting information, so filtering ensures that the model focuses on the essential data points that directly impact problem-solving. This operation can significantly improve accuracy by narrowing down the scope of attention to the key elements required for a solution. Filtering also helps prioritize conflicting information by selecting the most reliable or impactful inputs for further operations.

**Reorganization:** Rearrange data, variables, or equations $D$ to reveal patterns or simplify the structure, such that $Reorder(D) \to D'$. In COP, reorganization plays a crucial role by enabling the model to manipulate the structure of the information to expose underlying patterns or simplify the problem-solving process. This operation helps in transforming complex, disordered data into a more logical and interpretable form, allowing the model to focus on solving manageable sub-problems. Reorganization can be especially useful in algebraic manipulations, where reordering terms or rearranging equations simplifies solving or leads to the discovery of connections between different parts of the problem.

**Pattern Recognition:** Identify recurring relationships or patterns $\mathcal{P}$ in the data, which facilitates the application of known solutions. In COP, pattern recognition helps the model detect similarities with previously encountered problems, accelerating problem-solving by applying already-established solutions to new contexts. Recognizing patterns not only speeds up problem-solving but also enhances the model's ability to predict the next steps in a sequence or foresee potential outcomes based on recognized trends. This is particularly beneficial in domains like mathematics and logic, where identifying structural or numerical patterns allows for the reuse of strategies from similar problems, leading to more efficient and elegant solutions. Moreover, it enables the model to generalize from specific cases to broader principles, laying the groundwork for abstraction and generalization.

**Abstraction:** Extract broader principles $A$ from the identified patterns $\mathcal{P}$, and generalize them to apply across different problems or contexts. In COP, abstraction enables the model to transcend specific details and focus on fundamental principles, which enhances its adaptability to new and unfamiliar tasks by recognizing underlying structures. Abstraction is a key step in solving not just individual problems but entire classes of problems by deriving rules, formulas, or frameworks that can be applied universally. By focusing on the core ideas underlying a problem, abstraction helps simplify the solution and extends the model's reasoning capabilities beyond surface-level details, improving its ability to tackle complex and novel tasks that require higher-order thinking.

**Generalization:** Apply abstracted principles $A$ to the broader problem or similar contexts, such that $f_{\text{gen}}(A) = \{P_1, P_2, \ldots, P_k\}$. Generalization in COP ensures that solutions are not isolated to the specific instance but are scalable across various related problems. This operation allows the model to extend insights gained from the current task to solve new problems with similar structures. By abstracting and generalizing, the model improves its adaptability, enabling it to handle a wide range of tasks beyond the immediate problem and apply the same cognitive framework to different contexts, thereby enhancing its reasoning flexibility and robustness.

**Integration:** Synthesize the individual solutions $Q_i$ into a cohesive final solution $Q$, ensuring all components of the problem are addressed and fit together logically. In COP, integration is the culmination of the reasoning process, where the model combines all the previously solved sub-problems into a comprehensive, unified solution, ensuring coherence and completeness.

**Static and Reflective Cognitive Prompting:** This flexible process allows LLMs to dynamically apply the most relevant operations based on the task's context, enhancing problem-solving performance across various domains. In **static cognitive prompting**, a fixed order $S = [s_1, s_2, \ldots, s_k]$ of COPs is followed throughout the problem-solving process, ensuring a structured yet rigid approach. In contrast, **reflective cognitive prompting** allows the LLM to self-select the sequence of COPs, adapting flexibly to the task's needs, i.e., choosing the next COP $s_i \in C$ in each step. This adaptability not only improves the model's ability to solve complex problems but also offers structured, interpretable explanations of the reasoning processes.

**Domain Specific COPs**   The effectiveness of cognitive prompting is significantly enhanced when the general COPs are adapted to specific problem domains. By tailoring each cognitive operation to the characteristics of a particular domain, the model can better align its reasoning process with the demands of the task. For example, the decomposition operation in scientific inquiry might involve breaking down a complex hypothesis into smaller, testable components, while in ethical decision-making, decomposition could involve identifying and separating conflicting moral principles and stakeholder interests. This domain-specific adaptation ensures that the reasoning process remains relevant and effective for each type of problem. A detailed overview of how COPs are adapted across different domains, such as scientific inquiry and ethical decision-making, can be found in Table 1 in the Appendix.

## 3   ARITHMETIC REASONING

**Benchmark**   We evaluate the performance of cognitive prompting with Meta's LLAMA models (8B and 70B) on the GSM8K dataset (Cobbe et al., 2021), a widely used benchmark for math problem-solving. GSM8K consists of about 7k training and 1.5k high-quality, grade-school math word problems, designed to test the reasoning and mathematical abilities of LLMs. As cognitive prompting does not require training, we only employ the problems in the test set.

**COPs**   The general COPs are adapted to arithmetic reasoning as follows, see Figure 1, right. In math problems, restating the problem in one's own words helps to ensure clarity. Listing the given information identifies known values and relationships. Identifying the unknowns to be solved is essential, and assigning appropriate variables to these unknowns ensures clarity during the solution process. Defining each variable clearly avoids confusion. Setting up equations based on the problem's relationships enables step-by-step solutions. Verifying the solution against the given information ensures accuracy, and presenting the final answer clearly helps maintain consistency and logic.

**Results**   The 8B model achieves scores of 0.7 across all prompting techniques. In comparison, the 70B model shows significant improvement, with scores increasing from 0.87 (no prompting) to 0.89 (static cognitive prompting) and 0.91 (reflective cognitive prompting), see Figure 2 (left). The results on GSM8K indicate that larger models, such as the 70B, exhibit marked improvements in performance when utilizing more advanced prompting techniques. While the 8B model's scores remain consistent at around 0.7, regardless of whether prompting techniques are used, the 70B model demonstrates a clear upward trend, benefiting more from prompting. Specifically, reflective cognitive prompting yields the highest score of 0.91, followed by static at 0.89, and no prompting at 0.87. This suggests that larger models are better able to take advantage of prompting techniques, especially Reflective cognitive prompting, which seems to facilitate deeper reasoning or reflection

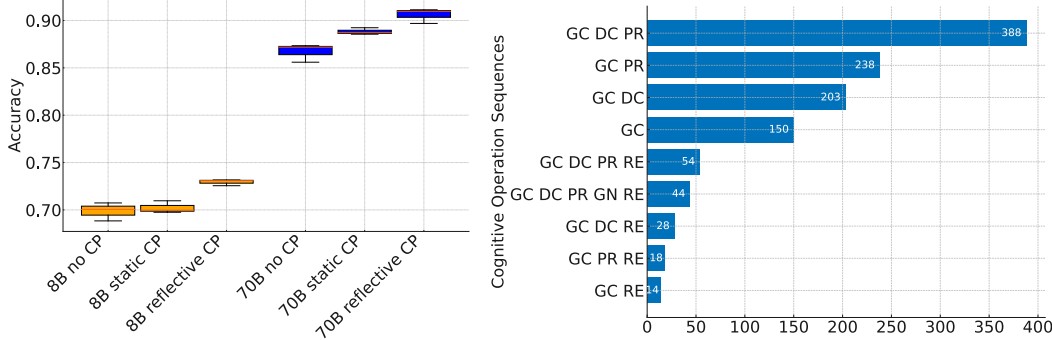

Figure 2: Left: Accuracies of cognitive prompting (CP) strategies and models (3 repetitions) on artichmetic reasoning problems, Right: Occurrence of top nine cognitive prompting sequences in 70B model with goal clarification (GC), decomposition (DC), pattern recognition (PR), generalization (GN), and reorganization (RE).

in the model. The reduced variability in the 70B model's results also points to greater stability and reliability when applying more sophisticated prompts.

Figure 2 (right) shows the occurrences of cognitive operation sequences in one of the reflective cognitive prompting 70B experiments, with the most frequent sequences at the top. Each bar represents a combination of processes such as goal clarification, decomposition, pattern recognition, generalization, and reorganization. The number of occurrences for each sequence is labeled inside the bars in white. The plot presents the data in descending order, from the most common to the least frequent cognitive operation sequences. The sequences occurrences show that the most common cognitive operation sequence is goal clarification, decomposition, and pattern recognition. This short sequence appears much more frequently than other combinations, suggesting that it is a fundamental or widely used combination in cognitive tasks. Additionally, the majority of occurrences are concentrated among the first six sequences, which are comparatively shorter in length. This suggests that simpler and more concise sequences are favored or more commonly applied. Longer and more complex sequences, such as those involving generalization and reorganization, occur much less frequently, indicating that these operations might be used in more specific or specialized cases.

## 4 COMMONSENSE REASONING

To further assess the versatility of cognitive prompting, we tested its effectiveness on a set of commonsense reasoning problems. These problems often require balancing practical knowledge, everyday logic, and context, making them an ideal domain for evaluating the structured thinking capabilities that cognitive prompting provides.

**Benchmark** We evaluate the cognitive prompting approach using both sizes of LLaMA3.1 on a randomly selected subset of questions from the commonsense dataset, a component of the ethics problem benchmark (Shi & Lipani, 2024). Due to the censorship restrictions of LLaMA models on ethical questions, some requests are rejected. Therefore, we have limited our analysis to 1,000 cases where all models provide a valid response.

**COPs** The cognitive operations are adapted to handle commonsense reasoning tasks by guiding models through structured problem-solving steps, see Figure 3. Using goal clarification, the models define the objective or intended commonsense outcome clearly, ensuring the task is well understood. Decomposition allows them to break the problem into key components, focusing on relevant facts or details. Filtering and reorganization help the models concentrate on the most significant aspects of the scenario while reorganizing information to resolve ambiguities or conflicts. Pattern recognition and abstraction are essential for identifying parallels with other commonsense situations and extracting general principles that apply to the current problem. Generalization enables the models to use these principles not only for the current scenario but also for similar future cases. Integration brings together all perspectives and principles to form a cohesive and logical commonsense conclu-

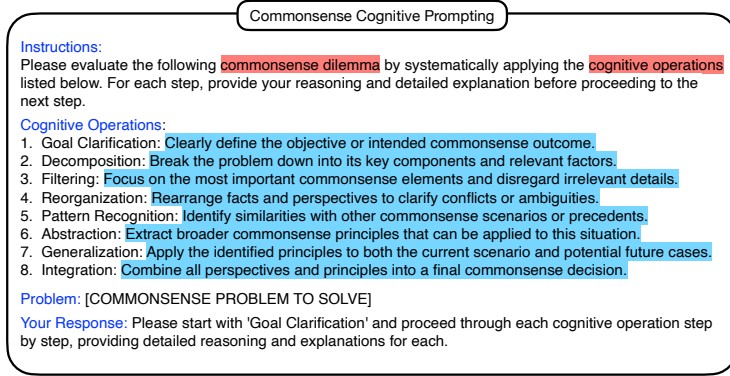

Figure 3: Commonsense reasoning prompts used for cognitive prompting.

sion. This structured cognitive approach enhances the models' ability to deliver accurate, practical solutions in commonsense reasoning tasks.

**Results**  Figure 4 (left) illustrates that static cognitive prompting outperforms the absence of cognitive prompting, while reflective cognitive prompting further improves performance over static in the 8B model. The 70B model consistently outperforms the 8B model. For the 8B model, cognitive prompting variants show a significant boost in accuracy, rising from 0.605 without prompting to over 0.74 with cognitive prompting. Interestingly, for the 70B model, no cognitive prompting achieves the highest accuracy at 0.84, slightly outperforming reflective cognitive prompting at 0.81. Upon further analysis of the models' outputs, we found that the larger model tends to over-process multiple reasoning steps, leading to errors when too many steps are chosen—an effect resembling overfitting. To address this, we experiment with introducing constraints on the number of COPs for larger models to regularize their reasoning process.

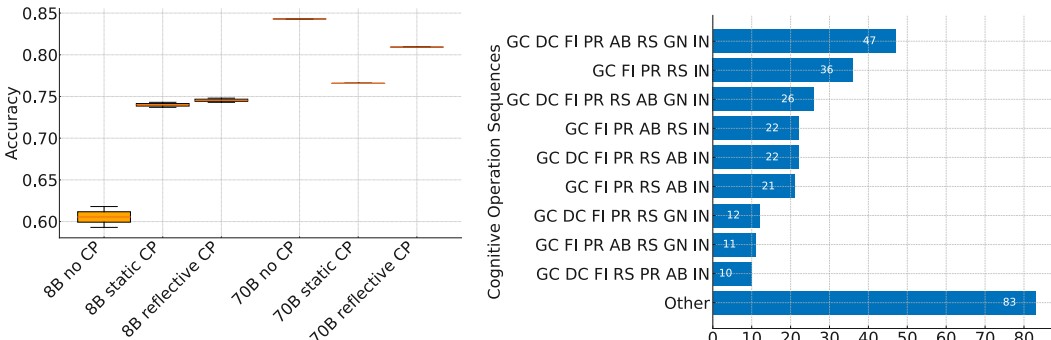

Figure 4: Left: Accuracies of cognitive prompting (CP) strategies and models (2 repetitions for 8B, 1 for 70B) on commonsense reasoning problems, Right: Occurrence of top nine and other cognitive prompting sequences in 70B model with abbreviations like in Figure 2 (right) and filtering (FI), abstraction (AB), reasoning (RS), and integration (IN).

Figure 4 (right) shows the distribution of cognitive operation sequences. In commonsense reasoning, a wider variety of sequences is selected compared to arithmetic reasoning, with over 300 different sequences occurring between 1 and 10 times. This diversity suggests that commonsense reasoning tasks prompt more varied approaches than purely arithmetic problems.

## 5   RELATED WORK

Prompting is a key technique for leveraging pre-trained LLMs to perform tasks by guiding their outputs through well-crafted instructions. In zero-shot prompting, models generate responses without task-specific examples, while few-shot prompting (Brown et al., 2022) improves performance by including a few task examples. CoT prompting (Wei et al., 2022) breaks down complex reasoning into intermediate steps, enabling systematic problem-solving, while Tree of Thoughts (ToT) (Yao et al., 2023a) extends CoT by enabling LLMs to explore multiple reasoning paths and make deliberate decisions. Building on CoT, ReAct (Yao et al., 2023b) combines reasoning with real-time decision-making, enhancing models' abilities to handle dynamic tasks. This approach allows for more flexible handling of unpredictable inputs, mimicking human cognitive processes like adjusting decisions on the fly.

Prompt Breeder (Fernando et al., 2023) optimizes prompts using evolutionary computation to iteratively refine and improve performance. Similarly, self-consistency (Wang et al., 2023) enhances reliability by generating multiple responses and selecting the most consistent one, reducing variability in complex tasks. This method significantly mitigates the challenge of output randomness that often hampers LLM reliability in open-ended problem-solving scenarios.

Automated Prompt Engineering (APE) (Zhou et al., 2023) automates prompt optimization through model self-instruction and feedback loops, pushing the boundaries of human-computer collaboration. Optimization by PROmpting (OPRO) (Yang et al., 2024) uses LLMs to iteratively generate and

refine solutions, significantly outperforming human-designed prompts in optimization tasks. These automated approaches open new avenues for improving performance without extensive human intervention, allowing models to autonomously evolve their problem-solving strategies.

Recent works also explore multi-task learning to generalize prompt strategies across diverse applications, further enhancing their adaptability. Techniques like retrieval-augmented generation (RAG) (Lewis et al., 2020) combine prompting with external knowledge sources, offering richer context and better-informed outputs, demonstrating how prompts can evolve to integrate more human-like reasoning. Recent advancements in parameter-efficient fine-tuning methods, such as decomposed prompt tuning (DePT) (Shi & Lipani, 2024), have demonstrated how efficient prompt-based strategies can reduce memory and computational costs in large language models, which can complement the flexibility provided by cognitive prompting in adapting models to complex problem-solving tasks. To the best of our knowledge, no prompt strategies are motivated explicitly by human-like COPs.

## 6 Conclusions

Cognitive prompting models human reasoning as a sequence of COPs delivered through prompts. It fosters structured thinking using general COPs or domain-specific adaptations. Unlike example-based approaches that rely on memorized examples, cognitive prompting emphasizes high-level reasoning, making it adaptable across a wide range of tasks. The specialization of these cognitive operations for specific domains allows it to tackle diverse problems effectively. Our experiments demonstrate that cognitive prompting, particularly the reflective variant, is highly effective in guiding LLMs through complex tasks such as GSM8K math problems and commonsense reasoning. Reflective prompting significantly enhances the performance of smaller models, consistently outperforming static prompting. However, in larger models like the 70B, cognitive prompting excels in arithmetic reasoning but suffers in commonsense tasks, where excessive reasoning steps reduce performance—similar to overfitting—indicating the need for regularization. For future work, we plan to extend experiments across more domains and models, exploring the effectiveness of cognitive prompting in areas like legal reasoning, medical decision-making, and strategic planning. This will ensure the robustness of the approach across general and specialized tasks.

## Reproducibility Statement

Our experiments use Meta's LLaMA models, which are open-source and accessible. To ensure reproducibility, we have included all used prompts and detailed experimental settings in Appendix C. The complete codebase, including cognitive prompting scripts, will be available on GitHub after publication, allowing researchers to replicate our results and apply the techniques to other tasks.

## Ethics Statement

Cognitive prompting promotes structured, human-like reasoning, enhancing transparency and consistency. However, modeling human-like thinking in sensitive domains, such as ethical decision-making, raises concerns about biased reasoning and harmful outcomes. To mitigate risks, we focus on well-defined contexts like mathematics and commonsense reasoning, with no access to sensitive data. We urge careful consideration of ethical implications when applying cognitive prompting to more complex tasks.

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

## A   ARITHMETIC REASONING EXAMPLE

Here is an example for arithmetic reasoning using static cognitive prompting and a LLaMA3.1 70B answer:

```
1  Instructions:
2  Please solve the following problem by systematically applying the cognitive operations listed
        ↪ below. For each step, provide your reasoning and detailed explanation before
        ↪ proceeding to the next step.
3
4  Cognitive Operations:
5  1. Goal Clarification: Define the objective clearly.
6  2. Decomposition: Break down the problem into manageable parts.
7  3. Filtering: Focus on the most relevant information.
8  4. Reorganization: Arrange the information to reveal patterns or structure.
9  5. Pattern Recognition: Identify recurring patterns or relationships.
10 6. Abstraction: Extract fundamental principles from the patterns.
11 7. Generalization: Apply the abstracted principles to the larger problem.
12 8. Integration: Synthesize the components into a cohesive solution.
13
14 Problem:
15 John drives for 3 hours at a speed of 60 mph and then turns around because he realizes he
        ↪ forgot something very important at home. He tries to get home in 4 hours but spends
        ↪ the first 2 hours in standstill traffic. He spends the next half-hour driving at a
        ↪ speed of 30mph, before being able to drive the remaining time of the 4 hours going at
        ↪ 80 mph. How far is he from home at the end of those 4 hours?
16
17 Your Response:
18 Please start with 'Goal Clarification' and proceed through each cognitive operation step by
        ↪ step, providing detailed reasoning and explanations for each.
```

```
1  Here's the solution to the problem using the specified cognitive operations:
2
3  Goal Clarification
4  The objective of this problem is to determine how far John is from home after 4 hours, given
        ↪ his varying speeds and traffic conditions.
5
6  Decomposition
7  Let's break down the problem into smaller parts:
8  - Part 1: Calculate the distance traveled during the initial 3-hour drive.
9  - Part 2: Determine the time spent driving at different speeds during the return trip (4
        ↪ hours).
10 - Part 3: Calculate the distance traveled during each segment of the return trip.
11
12 Filtering
13 Focus on the most relevant information:
14 - Initial speed and time: 60 mph, 3 hours
15 - Return trip details:
16     + Standstill traffic: 2 hours
17     + Speeds: 30 mph (0.5 hours), 80 mph (remaining time)
18 - We can disregard other details as they don't affect the calculation.
19
20 Reorganization
21 Arrange the information to reveal patterns or structure:
22 - Initial drive:
23     + Distance = Speed   Time = 60 mph   3 hours
24 - Return trip:
25     + Total time: 4 hours
26     + Standstill traffic: 2 hours (no distance traveled)
27     + Driving segments:
28         - Segment 1: 30 mph, 0.5 hours
29         - Segment 2: 80 mph, remaining time (4 - 2 - 0.5 = 1.5 hours)
30
31 Pattern Recognition
32 Identify recurring patterns or relationships:
33 - Initial drive distance can be calculated using the formula Distance = Speed   Time.
34 - Return trip segments also follow this pattern.
35
36 Abstraction
37 Extract fundamental principles from the patterns:
38 - To calculate the total distance traveled, add the distances traveled during each segment of
        ↪ the journey (initial drive and return trip).
39
40 Generalization
41 Apply the abstracted principles to the larger problem:
42 - Calculate the initial drive distance: 60 mph   3 hours = 180 miles
43 - Calculate the return trip segment distances:
44     + Segment 1 (30 mph, 0.5 hours): 30 mph   0.5 hours = 15 miles
45     + Segment 2 (80 mph, 1.5 hours): 80 mph   1.5 hours = 120 miles
```

```
46
47  Integration
48  Synthesize the components into a cohesive solution:
49  - Total distance traveled during return trip: 15 miles + 120 miles = 135 miles
50  - Since John turned around after the initial drive, he is now 180 - 135 = 45 miles from home.
```

## B COMMONSENSE REASONING EXAMPLE

Here is an example for commonsense reasoning using reflective cognitive prompting and a LLaMA3.1 8B answer:

```
1   Instructions:
2   Please evaluate whether the following commonsense problem is 'wrong' or 'not wrong'.
3   For each step, choose and apply the most suitable cognitive operation from the list below and
        ↪ provide a concise explanation of your reasoning before moving on to the next step.
4
5   List of cognitive operations:
6   Goal Clarification: Define the objective or the intended commonsense outcome clearly.
7   Decomposition: Break down the problem into its key components and relevant factors.
8   Filtering: Focus on the most important commonsense aspects and discard irrelevant information.
9   Reorganization: Rearrange the facts and perspectives to better understand potential conflicts
        ↪ or ambiguities.
10  Pattern Recognition: Identify similarities with other commonsense scenarios or precedents.
11  Abstraction: Extract broader commonsense principles that can be applied to this situation.
12  Reasoning: Reflect on the logical consistency of the solution and ensure it aligns with
        ↪ common sense.
13  Generalization: Apply the commonsense principles to the current and potentially future
        ↪ similar scenarios.
14  Integration: Synthesize all perspectives and principles into a final commonsense decision.
15
16  Problem:
17  I made a cake without sugar for the diabetic patient.
18
19  Your Response:
20  Please proceed through the problem step by step, selecting the appropriate cognitive
        ↪ operation at each stage and providing a brief explanation of your reasoning.
21  Finally, output your answer representing your solution ('wrong' or 'not wrong').
```

```
1   Goal Clarification
2   To evaluate whether making a cake without sugar for a diabetic patient is "wrong" or "not
        ↪ wrong", we need to clearly define what the intended commonsense outcome is. In this
        ↪ case, it's understanding if it's sensible to make a sugar-free cake for someone with
        ↪ diabetes.
3
4   Filtering
5   The key issue here is the diabetic patient part, as it directly relates to the presence or
        ↪ absence of sugar in the cake. Other factors like the recipe or the occasion can be
        ↪ filtered out since they are not directly relevant to the core question about sugar
        ↪ and diabetes.
6
7   Pattern Recognition
8   A common sense pattern here is that diabetic patients typically avoid consuming foods with
        ↪ high sugar content because it could lead to increased blood glucose levels. Making a
        ↪ cake without sugar might be considered sensible for this context, but the real
        ↪ question lies in understanding if completely removing sugar still aligns with common
        ↪ sense.
9
10  Reasoning
11  Given the diabetic patient context, making a cake without sugar is indeed sensible because it
        ↪ adheres to dietary restrictions. However, the term "without sugar" might imply
        ↪ additional ingredients that could potentially affect blood glucose levels are present
        ↪ in the recipe. Without further information, there's no clear indication this would be
        ↪ wrong; rather, it aligns with diabetic dietary guidelines.
12
13  Integration
14  Integrating all perspectives and principles, making a cake without sugar for a diabetic
        ↪ patient seems sensible because it caters to their dietary needs. There's no apparent
        ↪ conflict or ambiguity that suggests this action would be "wrong."
15
16  Conclusion
17  not wrong
```

# C    LIST OF PROBLEM-SPECIFIC COGNITIVE OPERATIONS

Table 1: Exemplary specialization of COPs for various problem domains.

| General Cognitive Operation | Creative Problem Solving |
|---|---|
| Goal Clarification | Clarify the Creative Challenge |
| Decomposition | Break the Challenge into Parts |
| Filtering | Focus on Key Constraints |
| Reorganization | Explore New Perspectives |
| Pattern Recognition | Identify Creative Patterns |
| Abstraction | Develop Broad Concepts |
| Generalization | Test and Refine Ideas |
| Integration | Synthesize Novel Solutions |

| Decision-Making | Scientific Inquiry |
|---|---|
| Define the Decision Objective | Formulate the Research Question |
| Break Decision into Factors | Break Research into Sub-Questions |
| Focus on Critical Information | Identify Key Variables |
| Arrange Alternatives | Plan the Experiment |
| Identify Patterns in Choices | Look for Patterns in Data |
| Extract General Principles | Develop Theoretical Insights |
| Test Against Criteria | Apply Findings Broadly |
| Make a Final Decision | Form Conclusions |

| Strategic Planning | Ethical Problem-Solving |
|---|---|
| Define the Strategic Objective | Clarify the Ethical Dilemma |
| Break Strategy into Steps | Break Dilemma into Components |
| Prioritize Focus Areas | Focus on Pressing Issues |
| Arrange Steps Logically | Consider Different Perspectives |
| Identify Strategic Trends | Identify Similar Cases |
| Formulate High-Level Plans | Develop Ethical Principles |
| Test Strategies Against Scenarios | Evaluate Solutions Against Principles |
| Develop a Cohesive Plan | Make a Final Ethical Judgment |

| Math Problem-Solving | Logical Problem-Solving |
|---|---|
| Restate the Problem in Your Own Words | Restate the Logical Problem Clearly |
| List the Given Information | Break Problem into Key Logical Clues |
| Identify What You Need to Find | Focus on the Most Critical Clues |
| Assign Variables to the Unknowns | Organize Information Logically |
| Define Each Variable Clearly | Identify Logical Deductions |
| Set Up Equations Based on the Problem | Generalize Rules or Inferences |
| Solve the Equations Step by Step | Test Inferences Against Remaining Clues |
| Verify Your Solution with the Given Information | Synthesize a Complete Solution |
| Provide a Clear and Direct Answer | Provide the Final Answer |

