# OpenReview forum: "Unlocking Structured Thinking in Language Models with Cognitive Prompting"
_ICLR.cc/2025/Conference — ICLR 2025 Conference Withdrawn Submission_

### Official Review · Reviewer_xtYj · 2024-10-29

**Soundness:** 2
**Presentation:** 2
**Contribution:** 1
**Rating:** 3
**Confidence:** 5

**Summary:**

This paper introduces a novel prompting method called cognitive prompting, in which a large language model is instructed to select an operation from a predefined operation list at each reasoning step. Experimental results on the GSM8K dataset and a commonsense reasoning benchmark demonstrate the effectiveness of this proposed prompting method.

**Strengths:**

- The approach of organizing problem-solving procedures through a structured sequence is intuitive and well-motivated. However, the method's generalizability needs careful examination.
- The experimental results indicate that designing a list of basic operations and allowing LLMs to select one at each reasoning step effectively enhances performance on certain tasks, such as the GSM8K dataset in mathematics. I would encourage the authors to further explore the potential benefits of this approach across diverse datasets and domains.

**Weaknesses:**

- The proposed method depends heavily on a human-defined operation list, which might limit the generalizability of the method.
- The evaluation lacks comprehensiveness, as it uses only one dataset for the arithmetic reasoning task and one benchmark for the commonsense reasoning task. The current experimental results are insufficient to substantiate the claim that "Unlike example-based approaches that rely on memorized examples, cognitive prompting emphasizes high-level reasoning, making it adaptable across a wide range of tasks." To provide a fair comparison with example-based approaches, such as few-shot CoT [1], additional experiments on other datasets should be considered, for example, those used in [1], [2], [3], and [4].
- Some experimental details are missing, e.g.,
  - The prompting method corresponding to "no CP" in Figure 2 is not described.
  - Detailed information about the evaluation dataset for the commonsense reasoning task is not provided.
- Potential errors in the content:
  - Line 190: the statement "The 8B model achieves scores of 0.7 across all prompting techniques. " appears incorrect, as "8B reflective CP" achieves an accuracy of approximately 0.73.

References
- [1] Wei et al., Chain-of-thought Prompting Elicits Reasoning in Large Language
  Models, NeurIPS 2022.
- [2] Kojima et al., Large Language Models Are Zero-shot Reasoners, NeurIPS 2022.
- [3] Yao et al., ReAct: Synergizing Reasoning and Acting in Language Models, ICLR 2023
- [4] Wang et al., Self-Consistency Improves Chain of Thought Reasoning in Language Models, ICLR 2023

**Questions:**

As mentioned before, the idea of organizing problem-solving procedures through a structured sequence is intuitive and well-motivated. However, the method's generalizability needs careful examination. I would encourage the authors to further explore the potential benefits of this approach across diverse datasets and domains.

---

### Official Review · Reviewer_DMge · 2024-11-01

**Soundness:** 3
**Presentation:** 3
**Contribution:** 2
**Rating:** 5
**Confidence:** 2

**Summary:**

The paper introduces an approach called cognitive prompting to enhance problem-solving abilities in large language models (LLMs) by mimicking human cognitive operations (COPs). These operations include goal clarification, decomposition, filtering, reorganization, pattern recognition, abstraction, generalization, and integration. The authors evaluate the effectiveness of this approach using LlamA models on arithmetic reasoning tasks from the GSM8K dataset and commonsense reasoning benchmarks. They compare models without cognitive prompting, models with a static sequence of cognitive operations, and models using reflective cognitive prompting where the LLM dynamically selects the sequence of operations. Results indicate performance improvements, especially with reflective cognitive prompting in larger models, demonstrating enhanced interpretability and flexibility in problem-solving.

**Strengths:**

- The experiments are conducted rigorously, with thorough comparisons and analyses demonstrating the effectiveness of the proposed approach.
- The paper is well-structured and clearly articulates the concept of cognitive prompting, its implementation, and its benefits.

**Weaknesses:**

Please see the questions

**Questions:**

Can the authors provide more detailed descriptions of the implementation of each cognitive operation and the dynamic selection process for reflective cognitive prompting?

How do the authors plan to address the issue of overprocessing/overthinking in larger models?

Are there any plans to validate the effectiveness of cognitive prompting across more diverse domains, e.g., RL decision-making?
For instance see: Momennejad, Ida, et al. "Evaluating cognitive maps and planning in large language models with CogEval." Advances in Neural Information Processing Systems 36 (2024).

---

### Official Review · Reviewer_8TwP · 2024-11-03

**Soundness:** 2
**Presentation:** 2
**Contribution:** 2
**Rating:** 3
**Confidence:** 4

**Summary:**

This paper proposes cognitive prompting to enhance the structured thinking capabilities of LLMs. The prompting method offers cognitive operations such as goal clarification, decomposition, abstraction, among others. Utilizing cognitive prompting, LLMs are capable of human-like thinking to simplify complex questions. This paper also evaluates cognitive prompting using benchmarks in arithmetic reasoning and commonsense reasoning. The results show that cognitive prompting can improve the performance of LLMs.

**Strengths:**

1. This paper is well-written. The explanation of each component in cognitive prompting is clear.
2. This paper makes LLMs equipped with cognitive operations, which is insightful for exploring the thinking modes of LLMs and future prompting design.
3. This paper exhibits various cognitive processes of different questions. In experiments, it counts the frequency of each combination of operations, which can reflect the internal thought process of LLMs.

**Weaknesses:**

1. The experiments are not comprehensive. Only two datasets are used in the paper. It would be better to have more results on datasets of varying difficulties (like MATH [1]).
2. The gain brought by cognitive prompting is minor and unstable. In arithmetic reasoning, the improvements are relatively limited. In commonsense reasoning, the use of cognitive prompting severely drops performance for the 70B model.
3. There is a lack of baselines. This paper only compares CP with the vanilla models, but there are also some prompting methods for enhancing reasoning capabilities (Chain of Thought[2], Tree of Thought[3], Reflexion[4]).
4. This method's effectiveness depends on the design of cognitive operations, which leads to poor generalization.

[1] Hendrycks D, Burns C, Kadavath S, et al. Measuring mathematical problem solving with the math dataset[J]. arXiv preprint arXiv:2103.03874, 2021.
[2]Wei J, Wang X, Schuurmans D, et al. Chain-of-thought prompting elicits reasoning in large language models[J]. Advances in neural information processing systems, 2022, 35: 24824-24837.
[3]Yao S, Yu D, Zhao J, et al. Tree of thoughts: Deliberate problem solving with large language models[J]. Advances in Neural Information Processing Systems, 2024, 36.
[4]Shinn N, Cassano F, Gopinath A, et al. Reflexion: Language agents with verbal reinforcement learning[J]. Advances in Neural Information Processing Systems, 2024, 36.

**Questions:**

Are these cognitive operations handcrafted? How does it transfer to other questions?

---

### Official Review · Reviewer_6EzR · 2024-11-04

**Soundness:** 3
**Presentation:** 3
**Contribution:** 1
**Rating:** 3
**Confidence:** 4

**Summary:**

In the paper, the authors suggest prompting LLMs based on human cognitive operations for guiding problem-solving.

**Strengths:**

Potentially interesting idea.

**Weaknesses:**

The paper outlines an interesting idea, however, it
1) does not sufficiently qualify major differences in the considered/realised key COPs, compared to related work, including the various contributions of in-context learning, instruction/demonstration finetuning, and CoT prompting. As an arbitrary example: How is filtering (FI) differently realised and investigated as in previous approaches?

Furthermore, the current version insufficiently studies the idea. The results seem too coarse for understanding and valuing significant differences in related work. In particular, the following points should be considered:
2) The quantitative results both in sec 3 and 4 need comparisons with baselines, e.g. finetuning and CoT prompting.
3) For the universality of the results, other models, both open and proprietary, need to be investigated as well, such as Mistral/Bloom and GPT/Gemni/Claude.
4) Qualitative results are missing, thus particular representative cases for good and bad accuracies of the key COPs.

Therefore, I do not see a significant and novel contribution to ICLR.

**Questions:**

none.

---

### Note · Authors · 2024-11-16

**Comment:**

Thank you very much for the reviews. We appreciate your effort and send the work somewhere else.

**Withdrawal Confirmation:**

I have read and agree with the venue's withdrawal policy on behalf of myself and my co-authors.